# Mechanism of Rice Resistance to Bacterial Leaf Blight via Phytohormones

**DOI:** 10.3390/plants13182541

**Published:** 2024-09-10

**Authors:** Qianqian Zhong, Yuqing Xu, Yuchun Rao

**Affiliations:** College of Life Sciences, Zhejiang Normal University, Jinhua 321004, China; zqq0220@zjnu.edu.cn (Q.Z.); yq9770423@163.com (Y.X.)

**Keywords:** resistance, bacterial leaf blight (BLB), phytohormones, regulation network, molecular breeding

## Abstract

Rice is one of the most important food crops in the world, and its yield restricts global food security. However, various diseases and pests of rice pose a great threat to food security. Among them, bacterial leaf blight (BLB) caused by *Xanthomonas oryzae* pv. *oryzae* (*Xoo*) is one of the most serious bacterial diseases affecting rice globally, creating an increasingly urgent need for research in breeding resistant varieties. Phytohormones are widely involved in disease resistance, such as auxin, abscisic acid (ABA), ethylene (ET), jasmonic acid (JA), and salicylic acid (SA). In recent years, breakthroughs have been made in the analysis of their regulatory mechanism in BLB resistance in rice. In this review, a series of achievements of phytohormones in rice BLB resistance in recent years were summarized, the genes involved and their signaling pathways were reviewed, and a breeding strategy combining the phytohormones regulation network with modern breeding techniques was proposed, with the intention of applying this strategy to molecular breeding work and playing a reference role for how to further improve rice resistance.

## 1. Introduction

Rice is one of the most important food crops in the world, and its yield restricts global food security [1]. However, various diseases and pests of rice pose a great threat to food security, such as plant viruses, bacteria, fungi, and some plant pests (*Nilaparvata lugens*, *Cnaphalocrocis medinalis*, and *Chilo suppressalis*) [2,3]. For example, bacterial leaf blight (BLB) caused by *Xanthomonas oryzae* pv. *oryzae* (*Xoo*) is one of the most serious bacterial diseases affecting rice in the world [4], which can cause 50% of rice yield in severe cases [5,6]. The disease is most prevalent in Asia, particularly in countries like India, China, and Japan [7]. *Xoo* usually enters the rice through water pores or wounds and then multiplies in vascular bundles to clog them. The disease typically first appears at the tip of the leaf and then gradually yellows and spreads throughout the leaf until it dies, resulting in a decline in both rice yield and quality [7,8] (Figure 1). Traditional control methods mainly involve spraying pesticides, but their effect on the targeted BLB is limited, the control effect is poor, they have a high planting cost, and they cause pollution damage to the environment [9]. In addition, after the use of bactericides, the targeted bacteria will develop resistance, making it more difficult to control BLB, which is not conducive to the sustainable development of agriculture [10]. The screening and cloning of disease-resistant genes and the analysis of the regulatory mechanism of disease resistance are the basis of disease resistance breeding.

Up to now, research on BLB resistance has achieved significant progress, with at least 47 resistance genes identified [11]. However, the regulatory mechanism of defense molecules against foreign pathogens has not been fully analyzed. Rice will trigger two layers of immune system in the face of foreign pathogen invasion, including pattern recognition receptor (PRR)-triggered immunity (PTI) and effector-triggered immunity (ETI) [12]. In turn, these immune receptors will activate a series of immune responses, including phytohormone signaling pathways [13]. Over the past two decades, an increasing number of rice researchers have discovered that phytohormones are widely involved in disease resistance [14], making it a focal point in rice disease resistance research. Well-known phytohormones such as auxin [15], abscisic acid (ABA) [16], ethylene (ET) [17], jasmonic acid (JA) [18], and salicylic acid (SA) [19] have been reported to be involved in plant disease resistance, serving as key factors in the underlying mechanism. Therefore, gaining a deeper understanding of the role of phytohormones in disease resistance, along with further analysis of rice defense mechanism, is essential for cultivating varieties with BLB resistance and even broad-spectrum resistance. Therefore, if we can use a molecular breeding strategy that links disease-resistant genes with phytohormone resistance networks, we could greatly promote the research process of BLB resistance in rice. Moreover, as a classic model crop, rice provides a foundation for understanding the plant disease resistance regulatory network more broadly. In this paper, we have summarized the role of phytohormones in rice BLB resistance over recent years, reviewed the genes involved and their signaling pathways, and summarized the progress made in BLB resistance research, with the goal of analyzing the regulatory network of phytohormones involved in BLB and applying this knowledge to molecular breeding, aiming to cultivate a new BLB-resistant germplasm. 

## 2. Role of Phytohormones in Rice Resistance to BLB

Significant progress has been made in understanding the role of phytohormones in rice resistance to BLB. Researchers have identified some genes that respond to both phytohormones and BLB, modulating BLB resistance by regulating phytohormones, thus outlining a regulatory framework of phytohormones in rice resistance to BLB (Figure 2).

### 2.1. Role of SA in Rice Resistance to BLB

SA is an emerging phytohormone favored by many researchers, widely recognized for its role in both biotic and abiotic stresses in plants [20]. In rice, both the increase in endogenous SA level and the application of exogenous SA can moderately improve resistance to pathogens, including BLB caused by *Xoo* [21,22]. In the study by Le Thanh et al. [21], treatment with exogenous SA reduced the severity of BLB by more than 38%, and the levels of most defense markers in the plants were significantly different from those in the control group. Similar results were reported earlier by Mohan Babu et al. [22]. The team treated rice with SA and observed that endogenous SA levels increased by approximately threefold, and increased levels of phenols and pathogenesis-related (PR) protein such as chitinase and β-1, 3-glucanase were detected in the plants. These results indicate that the application of SA effectively triggers the defense response in rice, enhancing resistance by increasing the levels of defense-related substances.

Numerous genes that respond to BLB enhance rice resistance by controlling SA levels or SA signal transduction. Among these, *NPR1* is a key regulator of systemic acquired resistance (SAR) in both *Arabidopsis* and rice, serving as a classic representative gene that illustrates the role of phytohormones in defense response [23]. The expression of *OsNPR1* is induced by SA, and its overexpression can significantly improve rice resistance to BLB. After entering the nucleus, OsNPR1 interacts with TGA transcription factor (TF) [23,24,25], leading to increased resistance to rice blast [26]. Additionally, the resistance pathway involving *OsNPR1* is also related to IAA and JA signaling [24,27]. These findings suggest that *OsNPR1* is a broad-spectrum disease resistance gene, potentially regulating the antagonistic interaction of many phytohormone signaling pathways. Overexpression of the *OsNPR1* homologous gene, *OsNPR3.3*, also enhanced resistance to *Xoo* and was also induced by defense-related substances such as ET, benzothiadiazole, and MeJA. Interestingly, the SA signal mediating rice resistance is involved in the *xa*5-mediated immune response to BLB, yet it operates independently of *OsNPR1* [24,28,29]. *OsSAHs* are SA hydroxylase genes that catalyze SA in vitro and have been shown to play a crucial role in SA metabolism and disease resistance [30]. The research team inoculated *Xoo* to transgenic lines that knocked out and overexpressed *OsSAH2* and *OsSAH3*, revealing that *OsSAH2* and *OsSAH3* negatively regulate the SA metabolic pathway and resistance to *Xoo*. Liu et al. [31] also demonstrated that SA hydroxylase genes *OsS5H1*, *OsS5H2*, and *OsS5H3* exist as negative regulators of *Xoo* resistance. Recent studies by Zeng et al. [32] further verified that *OsS5H1* negatively regulated rice resistance to *Xoo* and also proved that the negative regulation of *OsMGT3* on BLB resistance is mediated by SA. *OsSGT1* encodes SA glucosyltransferase, which is involved in SA metabolism and can catalyze the conversion of SA to SA O-β-glucoside (SAG). Its transcription is induced by *Xoo* and can be activated by *OsTGAL1*, resulting in a negative regulation of *Xoo* resistance [33,34]. *OsACBP5* is a positive regulator of broad-spectrum resistance in rice, and its resistance to *Xoo* is SA-dependent [35]. *ROD1* can develop broad-spectrum resistance to a variety of pathogens after loss-of-function, including *Xoo*. At the same time, the deficient mutation accumulated higher SA and JA levels in the plant, thereby conferring stronger resistance to BLB [36]. The gene also plays a unique role in balancing rice growth and defense. All of the aforementioned genes are involved in resistance to *Xoo* through SA. 

Recent studies on the involvement of SA in rice resistance to *Xoo* have made some progress, uncovering shared mechanisms. For example, MAPK signal transduction plays a crucial role in the immune response mediated by disease resistance (R) genes that are associated with SA signaling [37]. *OsMPK6* encodes a mitogen-activated protein kinase, which negatively regulates resistance to *Xoo*, and the enhancement of resistance is associated with the accumulation of ROS, SA, JA, and the upregulation of SA and JA signal transduction genes [38,39]. *OsMPK15* is also involved in the defense response of MAPK, and its knockout mutant has similar changes to *OsMPK6*, showing increased resistance to BLB in rice but a significant decrease in yield. The opposite trend was observed in the overexpressed mutants, which showed increased susceptibility to *Xoo* and yield. These results suggest that *OsMPK15* negatively regulates disease resistance by modulating SA and JA signaling pathways, favoring yield in the trade-off between disease resistance and yield [40]. NAC family members have been shown in many studies to participate in defense responses via phytohormones signaling [41,42]. Recent studies by Zhong et al. [43] have shown that OsNAC2 negatively influences the defense response to *Xoo* by interacting with OsEREBP1 and inhibiting the accumulation of SA levels and the expression of SA biosynthesis genes. *OsNAC2* has also been previously associated with phytohormone signaling pathways such as ABA, IAA, and ET [44,45,46]. WRKY TFs are proteins that regulate gene transcription and are widely involved in various physiological processes of plants. Certain WRKY genes are involved in regulating resistance to BLB by responding to biotic stress and modulating SA level [47]. In the study by Ryu et al. [48], it was found that one-third of the 45 WRKY genes tested in rice showed a significant response to *Xoo* inoculation. A pair of alleles, *OsWRKY45-1* (from japonica rice) and *OsWRKY45-2* (from indica rice), were previously identified as WRKY TF genes that mediate phytohormone response to *Xoo*. These alleles encode proteins with 10 amino acid differences and play opposite roles in resistance to BLB, likely due to their involvement in different defense signaling pathways. *OsWRKY45-1* is a negative regulator of *Xoo*; the resistance of *OsWRKY45-1* knockout lines increases with the accumulation of SA and JA. On the contrary, *OsWRKY45-2* only involves the regulation of JA [4]. Interestingly, the study by Shimono et al. [49] found that overexpressing *OsWRKY45* (from japonica rice), which corresponds to *OsWRKY45-1* in the study by Tao et al. [4], enhances rice resistance to *Xoo*. The inconsistency between these two studies may be due to differences in the overexpression methods used, leading to variations in gene expression patterns. Overexpression of *OsWRKY62* affects *xa21*-mediated resistance to *Xoo* and is a negative regulator of BLB [50,51], with its expression induced by SA [48]. On the contrary, overexpression of *OsWRKY6* can activate the expression of *OsICS1* and some pathogenesis-related genes, such as *OsPR10a*, thus showing increased resistance to *Xoo*. *OsICS1* participates in the synthesis of SA, suggesting that *OsWRKY6* regulates the rice defense response against BLB by influencing SA metabolism [52]. Similar to *OsWRKY6*, *OsWRKY13* has been shown to mediate resistance to *Xoo* by directly or indirectly regulating SA- and JA-dependent signaling pathways, while its overexpression can also enhance resistance to BLB [53]. *OsWRKY13* can also regulate responses to biotic stress from *Xoo* and drought stress by selectively binding to different cis-elements [54]. 

The involvement of SA in rice defense response usually depends on the change in SA level and the regulation of its signal pathways (Table 1). Overall, rice resistance to BLB is proportional to the level of SA, with the change in resistance usually accompanied by alterations in defense-related substances and defense-related genes. Many related genes control rice resistance by influencing SA metabolism. In addition, substantial evidence shows that the involvement of SA in resistance to BLB is usually accompanied by changes in JA level, but the mutual regulatory mechanism between the two needs to be further analyzed. 

### 2.2. Role of JA in Rice Resistance to BLB

JA is a lipid-derived phytohormone synthesized by plants. It is widely involved in many physiological phenomena in plant growth and development and is actively involved in responding to both biotic and abiotic stresses and thus endows plants with broad-spectrum resistance [55,56]. Infection by foreign pathogens and other forms of biotic stress can stimulate the rapid synthesis and accumulation of JA and its derivatives, subsequently promoting the expression of defense-related genes and the production of defense-related metabolites [57]. Examples include the diterpenoid phytoalexin momilactones and the flavonoid phytoalexin sakuranetin [58,59]. Multiple studies by Kenji Gomi’s team have confirmed this claim, finding that JA can induce the accumulation of a series of terpenoids that inhibit the spread of *Xoo*, thereby conferring resistance to BLB in rice [60,61,62]. 

The effect of JA in disease resistance has been well studied in dicotyledonous plants, but its role and mechanism in conferring disease resistance in monocotyledonous plants (such as rice and maize) remain not fully understood [56]. *OsHPL3* can mediate the specific defense response in rice, and its dysfunctional mutant *hpl3-1* exhibited a lesion mimic phenotype. The SA and JA levels of *hpl3-1* and wild type were detected and found that *hpl3-1* had a higher content on average; thus, its resistance to *Xoo* was stronger. Furthermore, the study of *OsHPL3* revealed that crosstalk between the hydroperoxide lyase (HPL) and allene oxide synthase (AOS) are branches of the oxylipin pathway, with *OsHPL3* playing a role in the activation of JA signaling, but the activation mechanism of SA signaling in *hpl3-1* has not yet been explored [63,64]. OsLOX10 and OsOPR10 are both key enzymes in the JA synthesis pathway, and both enzymes can positively regulate the expression of JA and SA signaling pathways in rice and contribute to resistance against BLB. This suggests that OsLOX10 and OsOPR10 may mediate both SA and JA signaling pathways in rice’s defense response [65,66]. *OsWRKY30* overexpression plants exhibit enhanced resistance to pathogens, indicating that this gene plays an important role in rice disease resistance. The transcription of *OsWRKY30* can accumulate rapidly in response to SA and JA, but *OsWRKY30* specifically stimulated the endogenous accumulation of JA [67]. This gene also plays an important role in OsMKK3-OsMPK7-OsWRKY30 [68] and OsMAPK6-OsLIC-OsWRKY30 [69] *Xoo* resistance modules. Deng et al. [70] reported a rice CCCH-type zinc finger protein C3H12 which is involved in rice-*Xoo* regulation. C3H12 acts as a nucleic acid-binding protein to positively regulate *Xoo* resistance. This regulatory process is associated with the accumulation of JA and the expression of JA signal transduction genes. Ke et al. [71] found that *OsPAD4*-RNAi plants showed increased susceptibility to *Xoo* and were accompanied by a decrease in JA accumulation, indicating that *OsPAD4*-mediated resistance to *Xoo* is JA-dependent. Interestingly, the team also found that *OsPAD4* is also involved in the defense response in *Arabidopsis*, but the response involved is SA-dependent. This suggest that *OsPAD4* may be involved in different signaling pathways mediated by defense response in different species. Yoo et al. [72] reported that the gene *rrsRLK*, which regulates ROS, acts as a negative regulator of resistance to *Xoo*. After infection by *Xoo*, *rrsRLK* loss-of-function mutant decreases ROS scavenging enzyme activity and thus accelerates the accumulation of H_2_O_2_, which leads to the upregulation of genes related to JA synthesis and PR genes such as *OsPR1a*. It may be a key regulator of ROS homeostasis. The lesion mimic gene *OsPHD1* encodes UDP–glucose–epimerase, the resistance of the lesion mimic mutant to *Xoo* is enhanced compared to the wild type, and the accumulation of JA and MeJA in the mutant is correspondingly increased. However, the expression of genes in the JA signaling pathway is downregulated in the mutant, and this regulatory mechanism could effectively prevent rice from over-defense responses [73].

JA-mediated resistance to *Xoo* in rice involves a complex regulatory network. The F-box protein CORONATINE INSENSITIVE 1 (COI1) is the main JA receptor in plants [55], and previous findings have confirmed the importance of COI1 protein in JA signaling, with the expression levels of more than 80% of JA response genes being decreased in *coi1* mutant [74]. *OsCOI1*-RNAi plants showed decreased sensitivity to JA and MeJA, while knockdown plants also showed decreased resistance to *Xoo*. In addition, *OsCOI1*-RNAi plants also showed typical hypersensitivity to the growth-related phytohormone gibberellic acid (GA) [75,76]. These findings suggest that this gene can correlate GA and JA signaling, providing preliminary evidence that plants prioritize defense over growth. Yamada et al. [77] reported that the JA ZIM-domain (JAZ) protein OsJAZ8 interacts with OsCOI1b in a coronatine-dependent manner in the same year. The team verified that the application of JA can enhance rice resistance to *Xoo* and found that OsJAZ8ΔC, a variant lacking the Jas domain, acts as a JA inhibitor to regulate rice resistance to *Xoo*. Hou et al. [78] reported a SAPK10-WRKY72-AOS1 module, where *WRKY72* transcription is induced by *Xoo* and JA, and it is a negative regulator of BLB. Under normal conditions, the SAPK10-mediated phosphorylation of WRKY72 weakens DNA methylation of the JA biosynthase gene *AOS1*, thereby maintaining normal JA levels. After infection by *Xoo*, the transcription of *SAPK10* is inhibited, and non-phosphorylated OsWRKY72 can affect endogenous JA synthesis and promote rice susceptibility. Additionally, the JA biosynthetase gene *OsAOS2* has also been shown to positively regulate resistance to BLB [79]. Uji et al. [80] reported a TF OsMYC2, which was shown to interact with multiple JA inhibitor OsJAZ proteins in a JAZ-interacting domain (JID) dependent manner. OsMYC2 is a positive regulator of resistance to *Xoo* and early JA response genes. RERJ1 has been shown to achieve a JA signaling-mediated defense response to *Xoo* by interacting with OsMYC2 [81]. The SA signaling regulatory protein OsNPR1 can interact with RERJ1 and inhibit its accumulation, thus inhibiting JA signaling and promoting SA signaling. Therefore, *RERJ1* is a key gene regulating SA and JA signaling antagonism [82]. Suzuki et al. [83] reported a MED25 subunit, where the loss-of-function mutant *osmed25* showed high resistance to *Xoo*, and OsMED25 negatively regulated part of OSMYC2-independent JA defense signaling. OsSRO1a, which interacts with JA signaling suppressor OsNINJA1, also interacts with OsMYC2. The overexpression of *OsSRO1a* exhibits the JA insensitive phenotype, resulting in decreased resistance to *Xoo*, indicating that *OsSRO1a* exists as a negative regulator of the OSMYC2-mediated JA signaling pathway [84]. More recently, the team reported that the MYB TF JMTF1, whose overexpression plants exhibit a JA-sensitive phenotype, enhances resistance to *Xoo* by promoting the accumulation of lignin and antibacterial monoterpene γ-terpinene [85]. The gene *OsCCD4b* reported by Taniguchi et al. [86] also enhances resistance to *Xoo* by inducing JA-related defense metabolites from JA. β-Cyc is a volatile degranulation carotenoid compound; its JA-induced accumulation level is regulated by OsJAZ8, and its synthesis is promoted by the OsCCD4b enzyme, which has antibacterial activity against *Xoo*. The above results provide key information for the analysis of JA’s involvement in the rice *Xoo* regulatory network, but its detailed mechanism has not been fully explored (Table 2). 

The involvement of JA in regulating rice BLB resistance usually depends on the interaction with SA, such as the classic genes *OsNPR1*, *OsWRKY45-1*, *OsMPK6*, *OsMPK615*, and *RERJ1*. Many studies have shown that the interaction between SA and JA exhibits different relationships in *Arabidopsis* and rice [87,88]. The latest study by Zhu et al. [89] further elucidates the synergistic effect of SA and JA in regulating BLB in rice. *OsEIL3* loss-of-function mutant weakens resistance to biotroph *Xoo*. Under normal conditions, *OsEIL3* confers resistance in rice by activating JA and SA biosynthesis after *Xoo* infection. Interestingly, infections by different pathogens activate different JA- and SA-dependent pathways. 


plants-13-02541-t002_Table 2Table 2JA-mediated BLB resistance and susceptible genes.GeneGene ProductPhytohormoneFunction CharacteristicRegulation ModeReference
*OsHPL3*
Hydroperoxide lyase JA, SAActivates JA-mediated defense responsesNegative[63,64]
*OsLOX10*
LipoxygenaseJA, SAUpregulates ROS levels and gene expression of the JA and SA signaling pathwaysPositive[65]
*OsOPR10*
12-oxygen-phytodienoate reductaseJA, SAUpregulates ROS levels and gene expression of the JA and SA signaling pathwaysPositive[66]
*OsWRKY30*
WRKY transcription factorJA, SAPromotes JA accumulation to improve resistancePositive[67]
*OsWRKY45-2*
WRKY transcription factorJAUpregulates JA level and the expression of defense-related genesPositive[4]
*C3H12*
CCCH-type zinc finger nucleic acid-binding proteinJAPositively regulates resistance to *Xoo* through the JA signaling pathwayPositive[70]
*OsPAD4*
Phytoalexin deficient 4JAParticipates in *Xoo* defense signaling via JA dependencyPositive[71]
*rrsRLK*
Cytoplasmic RLK JARegulates ROS levels and JA-related gene expressionNegative[72]
*OsPHD1*
UDP-glucose epimerase JADisruption of *OsPHD1* upregulates defense-related genes to activate the defense responseNegative[73]
*OsCOI1*
JA receptorJARegulates JA and GA, which jointly mediate rice disease resistancePositive[75,76]
*OsJAZ8*
JA ZIM-domain proteinJANegatively regulates JA-induced resistance to *Xoo*Negative[77]
*OsWRKY72*
WRKY transcription factorJA, ABAInhibition of *AOS1* transcription reduces JA levels, resulting in susceptibility to *Xoo*Negative[78]
*OsSAPK10*
Stress-activated protein kinaseJA, ABAPhosphorylates WRKY72, alleviating the inhibition of JA synthesisPositive[78]
*OsAOS1*
Allene oxide synthaseJAIncreases endogenous JA levels and thus improves resistancePositive[78]
*OsAOS2*
Allene oxide synthaseJA, SAMay mediate resistance by influencing JA levelsPositive[79]
*OsMYC2*
JAZ-interacting transcription factorJAThe resistance is mediated through JID-dependent interaction with the OsJAZ proteinPositive[80]
*RERJ1*
JA-inducible transcription factorJA, SAInteracts with OsMYC2 and OsNPR1 to regulate hormonal signalingPositive[81]
*OsMED25*
Mediator subunit 25JA, AuxinNegatively regulates a subset of OsMYC2-independent JA defense signalingNegative[83]
*OsSRO1a*
Similar to RCD one1aJANegatively regulates the OsMYC2-mediated JA signaling pathwayNegative[84]
*OsJMTF1*
MYB transcription factorJA, IAAPromotes the biosynthesis of lignin and antibacterial monoterpene, γ-terpinenePositive[85]
*OsCCD4b*
9-cis-epoxycarotenoid dioxygenaseJA, ABAPromotes the synthesis of β-Cyc and downregulates ABA levelsPositive[86]
*OsEIL3*
Ethylene-insensitive3-like3 transcription factor JA, SA, ETActivates JA and SA biosynthesis and signaling pathways to regulate resistancePositive[89]


### 2.3. Role of ET in Rice Resistance to BLB

ET is a common phytohormone involved in regulating various plant growth and development activities, including seed germination, seedling growth, organ development, and fruit ripening [90,91]. At the same time, ET is also an important phytohormone in response to external stress, with its mediated signaling pathway playing a significant role in rice disease resistance and defense response [92]. Studies have shown that the role of ET as a positive or negative regulator of disease resistance depends on the type of foreign pathogen [93]. Therefore, the analysis of the regulatory role of ET after infection by different pathogens can further understand the complexity of ET in plant defense response. 

In recent years, increasing evidence shows that ET can be a key factor involved in rice resistance to *Xoo*. Moon et al. [94] found that *Xoo* can induce the expression of *Oryza*-specific orphan gene *Xio1* in an immune receptor XA21-dependent manner, and ET and SA can also induce the expression of *Xio1*. Overexpression of *Xio1* in plants produced significant accumulation of ROS and H_2_O_2_ and showed enhanced resistance to *Xoo*. The EIN3/EILs TFs family is central to the ET signaling pathway and plays an important role in plant development and stress response [95]. Zhu et al. [96] found a homologous protein of EIN3/EIL OsEIL4, whose expression was upregulated 30-fold after *Xoo* induction. OsEIL4 positively regulates the expression of ET synthesis-related genes and defense-related genes, which is manifested as increased resistance to *Xoo*. The crosstalk between JA and SA is fine-tuned by the ET pathway, which plays an important role in regulating rice resistance to *Xoo*. The report of Zhu et al. [89] on *OsEIL3* reflects this crosstalk mode, which activates SA- and JA-mediated defense responses. *OsEDR1* can induce the synthesis of aminocyclopropane carboxylic acid (ACC) and the expression of the aminocyclopropane carboxylic acid synthase (ACS) gene family to promote the accumulation of ET and then inhibit the expression of genes related to SA and JA, resulting in decreased resistance to *Xoo* in rice and alleviated the appearance of lesions [97]. However, this result is contrary to the effect of many ET regulating resistance, indicating that its regulatory mechanism remains to be further analyzed. 

Previous studies have shown that ET plays an important role in the regulation of resistance to BLB in rice (Table 3). In particular, the rice resistance response co-mediated by ET, SA, and JA can resist the infection of pathogens through complex signaling networks. However, there are still few studies on the resistance of ET to BLB, and the role of ET is not conservative. Therefore, it is of great significance to further explore the role of the ET-mediated signaling pathway in the resistance mechanism of BLB. 

### 2.4. Role of Auxin in Rice Resistance to BLB

Auxin is an important phytohormone, and indole acetic acid (IAA) is the major form of it, which is a key factor in the resistance mechanism of rice [98]. Many pathogenic microorganisms can use tryptophan as a precursor to synthesize auxin, such as *Magnaporthe oryzae* (*M.oryzae*) [15] and *Xoo* [99]. Many existing studies have shown that auxin usually exists in a negative way to regulate BLB [98]. The plant cell wall is an important part of the basic resistance of rice, and auxin can induce the expression of extensin and loosen the cell wall, thus making the plants more susceptible to pathogen infection [99,100]. 

GH3 auxin-amido synthetases, encoded by *GH3* gene family members, can catalyze excess phytohormones in plants to change from the free state to amino acid conjugation, thus controlling the homeostasis of endogenous phytohormones [101]. Overexpression of *OsGH3.8* can catalyze the conversion of free IAA to IAA-Asp, preventing the accumulation of IAA, thus improving BLB resistance. The expression of this gene is induced by auxin, SA, and JA, but it only depends on the regulation of auxin on the cell wall and is independent of the SA- and JA-mediated signaling pathways [99]. At the same time, *OsGH3.8* can also be upregulated by *OsNPR1* to disrupt the auxin pathway and thus affect rice growth and development without impacting disease resistance [27]. The study reveals a link between SA and auxin in regulating defense and growth mechanisms, known as the “growth-defense trade-off”, providing more evidence that plants often prioritize defense over growth when these pathways intersect [76,102]. Similar to *OsGH3.8*, the overexpression of *OsGH3.1* and *OsGH3.2* can also enhance rice resistance to pathogens by reducing endogenous IAA levels [15,103]. The above three genes belong to the *GH3* family group II, with similar substrate preferences and targeted auxin catalysis [104]. However, *OsGH3.1* primarily confers specific resistance to rice blast, while *OsGH3.2* is manifested as broad-spectrum resistance to both rice blast and BLB [15,103]. 

Moreover, in addition to research on the *GH3* gene family, many studies have demonstrated that IAA is involved in regulating rice resistance to BLB. The cytochrome P450 gene *CYP71Z2* has been shown to confer resistance to *Xoo* in rice by regulating the biosynthesis of diterpene phytoalexins [105,106]. Further studies on this gene’s resistance mechanism revealed that it mediated rice resistance to *Xoo* by inhibiting the accumulation of IAA and the expression of extensin genes, independent of the common SA and JA signaling pathways involved in rice resistance to BLB [107]. There is also evidence that the resistance gene to *Xoo OsMED25*, which is related to JA signaling, is also involved in auxin signaling [83]. OsMED25 interacts with some auxin response factors (OsARFs) that regulate auxin, and the loss-of-function mutant *osmed25* did not show susceptibility to *Xoo* after IAA treatment. It was found that mutated OsMED25 lost its interaction with OsARFs, which led to the downregulation of auxin-related gene expression in the mutant, and thus showed an auxin-insensitive phenotype in *Xoo* resistance. Similarly, Uji et al. [85] recently reported the regulation of *Xoo* by MYB TF *JMTF1*, whose overexpressed transgenic plants were sensitive to JA and had increased resistance, while also exhibiting typical auxin-related phenotypes such as deficient gravity response and reduced lateral root number. Further studies showed that *JMTF1*-mediated JA signaling regulates resistance to *Xoo* by upregulating the expression of the auxin signal transduction inhibitor *OsIAA13*, thereby coordinating the crosstalk between defense and growth in rice. In many model plants, small RNA play an integral role in defense responses [108,109], mostly involving auxin signal transduction [110]. Zhao et al. [111] detected the expression profile of small RNA in rice leaves after *Xoo* infection and found that 60% of these small RNA were involved in the auxin signaling pathway in response to BLB. In previous studies, *miR1432* has been shown to increase auxin content by inhibiting its own expression, thereby regulating the grain filling rate and enhancing rice yield [112]. Recent reports have shown that *miR1432* mediates disease resistance by targeting the rice BLB negative regulatory gene *OsCaML2*. Overexpression of *miR1432* can improve resistance to *Xoo*, while *OsCaML2* overexpression transgenic plants show susceptibility to *Xoo* infection [113]. Gu et al. [114] reported an OSSGS3-tasiRNA-OSARF3 module, where *OsARF3* is involved in the regulation of auxin, and the negative regulator of *Xoo OsSGS3* can regulate the biosynthesis of tasiRNA that targets *OsARF3*. This module can also regulate the resistance to BLB and heat resistance of rice. Along with the study by Vemanna et al. [115], the crosstalk trade-off between biotic and abiotic stresses was confirmed. 

At present, some progress has been made in the study of auxin in BLB resistance (Table 4). Auxin usually promotes rice susceptibility to *Xoo* and co-regulates with other phytohormones. However, the detailed regulatory mechanism has not been fully explored and requires further analysis. 

### 2.5. Role of ABA in Rice Resistance to BLB

ABA, as a phytohormone, can regulate almost all plant growth and development processes, including seed dormancy, germination, flowering, and senescence [116]. It has also been found to play a widespread role in plant defense responses [117,118], with its role as a positive or negative factor depending on the type of invading pathogen [98]. Against *Xoo* invasion, ABA usually plays a negative role [119,120]. 

Compared to the abundant research on common defense-related phytohormones, ABA has been less discussed as a defense-related factor. Known studies have shown that pathogens often mediate defense responses through ABA by regulating its biosynthesis or affecting its signal transduction pathways, and that ABA itself interferes with plant defense pathways [120]. ABA also regulates disease resistance by controlling stomatal opening [121]. Both SA and JA usually exist as positive regulators of rice BLB. Xu et al. [120] demonstrated that ABA is a negative regulator of *Xoo*, with its immunosuppressive effect partly mediated by inhibiting SA-mediated defense pathways. β-Cyc, which is JA-induced and synthesized by the OsCCD4b enzyme, can downregulate the expression of ABA-responsive genes, thereby reducing ABA levels and enhancing resistance [86]. As reported by Yamaguchi et al. [122], *OsPLDβ1* knockdown transgenic plants accumulate ROS even without pathogen infection and spontaneously activate the defense response, and inhibition of *OsPLDβ1* expression will show an enhanced BLB resistance phenotype. In the study by Li et al. [123], inhibition of *OsPLDβ1* expression can also reduce sensitivity to exogenous ABA, so *OsPLDβ1* is a negative regulator of BLB. The study by Lu et al. demonstrated that *OsCPK24* is associated with the ABA signaling pathway. A bionic pesticide, ethylicin, can inhibit the growth and development of *Xoo* by disrupting cell morphology. Additionally, ethylicin can affect the activity of defense enzymes in the ABA signaling pathway to induce the expression of *OsCPK24*, ultimately enhancing rice resistance to *Xoo* [124]. 

Many TFs, such as WRKY, NAC, and bZIP, can mediate plant defense responses by regulating ABA signaling pathways [125]. In the SAPK10-WRKY72-AOS1 module reported by Hou et al. [78], the SAPK10-mediated phosphorylation of WRKY72 has long been shown to participate in ABA signal transduction. This study preliminarily elucidated the signaling pathway through which WRKY TFs regulate JA-ABA-mediated resistance to rice BLB. As reported by Son et al. [126], *OsWRKY114* is shown to be a negative regulator of ABA-induced *Xoo* susceptibility, and *OsWRKY114* enhanced resistance to *Xoo* by weakening the ABA inhibition of the SA signaling pathways and downregulating ABA biosynthesis and ABA-related genes. *OsWRKY76* has also been shown to be involved in the regulation of BLB in rice, where ABA treatment can induce its expression, negatively affecting *Xoo* resistance [50,127,128]. Liu et al. [129] revealed the regulatory mechanism of NAC TFs in ABA-mediated resistance to rice BLB. *ONAC066* positively regulates rice resistance to BLB. A yeast one-hybrid experiment confirmed that *ONAC066* can directly bind to the promoter of ABA-related genes *LIP9* and *NCED4*, enhancing disease resistance by regulating the ABA signaling pathway and promoting the accumulation of soluble sugar and amino acids. bZIP TFs can also regulate ABA-mediated resistance to BLB. The induction of the bZIP TFs *OsTFX1* depends on the type III effector gene *pthXo6*, which can increase rice susceptibility to BLB [130]. Liu et al.’s [131] study further found that *OsTFX1*/*bZIP71* is induced by ABA and that it is also involved in response to low-temperature stress. 

In general, previous studies on the role of ABA in rice BLB resistance have made preliminary progress (Table 5), but the signaling pathway of ABA-induced BLB susceptibility remains elusive and requires further exploration.

## 3. Mutual Regulatory Mechanism of Phytohormones in Rice Resistance to BLB

Given the long co-evolution between rice and *Xoo*, a relatively complex regulatory strategy has evolved. Each phytohormone activates a specific defense mechanism, and these different phytohormone pathways are integrated into a complex network of synergistic or antagonistic interactions known as phytohormone crosstalk [132]. When attacked by pathogens, rice develops a mix of phytohormones that coordinate to trigger defense responses [133]. 

In dicotyledonous plants, SA-JA is usually involved in defense responses through antagonism [134]. However, the role of SA-JA in response to pathogen invasion in monocotyledonous plants remains unclear. Tamaoki et al. [88] found that more than half of the genes induced by the SA analog benzothiadiazole (BTH) are also upregulated by JA. This suggests that JA and SA may interact synergistically in the transcriptional regulation of most rice defense genes. Hou et al. [13] further supported this statement. Among them, *OsNPR1* [24] and *OsMYC2* [80,81,82] are key players in SA-JA crosstalk and can activate different SA-JA interactions. Further studies on their molecular basis will provide deeper insights into the regulatory mechanisms of SA-JA crosstalk against *Xoo* resistance. In addition, many studies have proved that SA-JA crosstalk is mediated by ET [135]. The study by Zhu et al. [89] provided evidence that SA and JA signaling pathways respond synergistically to *Xoo* infection, with the immune response mediated through the EIN3 TF in the ET signaling pathway, and it is preliminarily speculated that EIN3 phosphorylation and histone modifications may affect the subsequent activation or inhibition of the SA and JA signaling pathways. This kind of crosstalk has already been reflected in the study of *OsEDR1* by Shen et al. [97]. JA is a positive regulator of *Xoo* resistance, while IAA typically contributes to susceptibility. Signaling pathways affecting JA-IAA crosstalk also exist in rice, such as JA-IAA crosstalk-mediated *Xoo* resistance involved in *OsMED25* [83] and *OsJMTF1* [86]. 

Additionally, there are many phytohormones that simultaneously regulate rice immunity and growth, with crosstalk between phytohormones balancing the relationship between growth and immunity. For example, *OsCOI1* [76] participates in GA-JA crosstalk, *OsNPR1* [27] participates in SA-IAA crosstalk, *JMTF1* [86] participates in JA-IAA crosstalk, etc. In rice, the crosstalk between these phytohormones basically prioritizes immunity over growth. Furthermore, there are many factors affecting phytohormone crosstalk, among which the relationship between biotic and abiotic stress is balanced through phytohormone crosstalk pathways affecting different leaf ages [136]. 

The aforementioned phytohormone pathways share many crosstalk centers, and various phytohormone crosstalks form a complex regulatory network that mediates resistance to *Xoo* (Figure 2), but the intricate relationships among these various pathways remain to be studied. The analysis of these phytohormone signaling pathways, especially the crosstalk centers, is crucial for designing *Xoo*-resistant rice varieties. 

## 4. Application of Modern Breeding Techniques in Cultivating Rice Resistant to BLB

Conventional breeding has played an important role in the development of BLB-resistant rice varieties, but it usually consumes a lot of manpower and time, and it is difficult to achieve the effect of multiple good genes’ polymerization [137]. An increasing number of BLB resistance genes in rice have been identified and cloned, driving the development and application of a series of modern breeding techniques in creating resistant germplasm. At present, widely used and effective modern breeding techniques include molecular marker-assisted selection (MAS) breeding, genetically modified breeding (GMB), and gene editing technology [138,139].

MAS involves selecting disease resistance traits by analyzing molecular marker genotypes closely linked with previously discovered resistance genes [140]. This molecular breeding method can aggregate one or more resistance genes into BLB susceptible varieties to achieve the effect of the aggregation of disease resistance traits and other excellent traits [137]. This approach is seen as a promising strategy for enhancing resistance to BLB [141]. As early as 2012, Ullah et al. [142] used this method to aggregate multiple BLB resistance genes without affecting the prized basmati aroma of aromatic basmati rice. Yugander et al. [143] also used this method to introduce BLB resistance genes *Xa21* and *Xa38* into APMS6B and retained the original excellent traits of the variety. Hairmansis et al. [144] adopted the same method to introduce the BLB resistance gene *Xa7* into the famous flood-tolerant variety Inpara 5 and brown plant hopper (BPH)-resistant variety Inpari 13. To some extent, this approach alleviates yield reduction in local rice varieties caused by various diseases. Therefore, the feasibility of transforming rice varieties with good agronomic traits but susceptible to BLB through MAS has been confirmed. But its application in BLB responses involving phytohormones has not yet been shown. 

GMB is another effective way to confer resistance to susceptible rice varieties. Both *Xa21* and *Xa23* are BLB resistance genes, and researchers have successfully transferred into several rice materials through transgenic means, and these materials have acquired BLB resistance [145,146]. This method can shorten the breeding cycle to some extent and is highly targeted. To date, numerous studies focusing on phytohormones’ regulation of rice resistance to BLB have already applied GMB. For example, research on *OsWRKY13* and *OsWRKY45-2*, both of which are involved in JA-mediated resistance to BLB in rice [4,53]. But the application of transgenic rice breeding for BLB resistance is still less common, possibly due to the lack of approval for transgenic rice. 

In recent years, gene editing technology has developed rapidly and has been gradually recognized for improving traits and increasing yield. In particular, CRISPR-Cas9-mediated genome editing introduces point mutations and provides a means to develop broad-spectrum resistance that maximizes resistance to pathogens [147]. *SWEET* genes belong to a class of sugar transport genes, and rice is susceptible to BLB when they are expressed. Researchers used CRISPR-Cas9 to edit the promoters of susceptibility genes *OsSWEET11*, *OsSWEET13*, *OsSWEET14*, and *OsSULTR3*, effectively reducing the BLB susceptibility of these genes [148,149]. Regarding the application of gene editing in phytohormone-mediated BLB resistance, there are also some precedents. For example, using CRISPR/Cas9 techniques to knock out *OsEDR1* and *OsMPK15* can produce lines with enhanced resistance, with both knockout lines showing the increased accumulation of SA and JA [40,150]. The CRISPR/Cas9 technique provides evidence for the involvement of phytohormones in rice defense responses to BLB. CRISPR/Cas9 technology undoubtedly provides an effective solution to the global food security crisis that may be faced by massive population growth.

The above methods provide a technical basis for breeding BLB-resistant rice varieties and can help us to obtain high-quality resistant varieties more quickly, efficiently, and accurately. 

## 5. Summary and Prospects

BLB is one of the most serious diseases affecting rice, significantly limiting its yield. Phytohormones effectively protect rice from the BLB, so exploring their regulatory mechanism is crucial for developing resistance to *Xoo*. Currently, there has been some progress made in the study of the phytohormones involved in rice immune responses to BLB. However, the detailed regulatory mechanism and the crosstalk between various phytohormones remain unclear and require further analysis.

With the discovery of more resistance genes and the mining of molecular markers, along with the development of molecular breeding techniques, MAS, GMB, and gene editing breeding have gradually become a trend and made great contributions to improving rice resistance. Understanding the molecular mechanisms of phytohormones regulation in BLB resistance can provide a theoretical basis for the cultivation of resistant rice varieties that balance growth and defense. Modern technologies are powerful tools that can turn this knowledge into reality, providing a new strategy for breeding resistant varieties in the future (Figure 3). 

## Figures and Tables

**Figure 1 plants-13-02541-f001:**
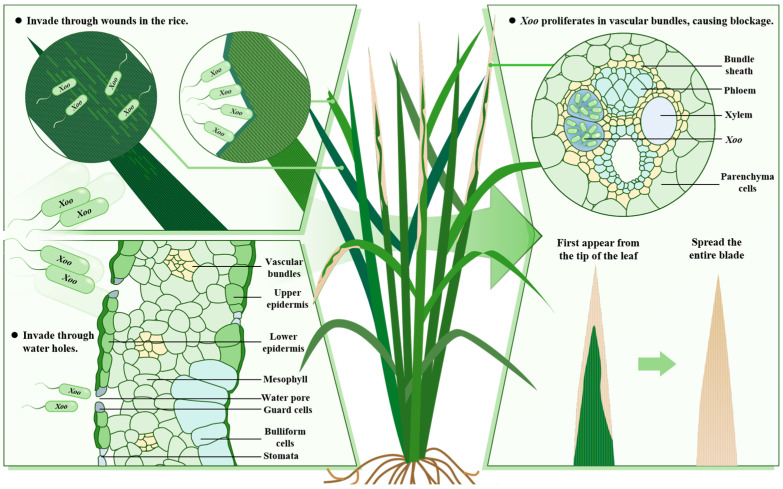
The process of *Xoo* infection of rice.

**Figure 2 plants-13-02541-f002:**
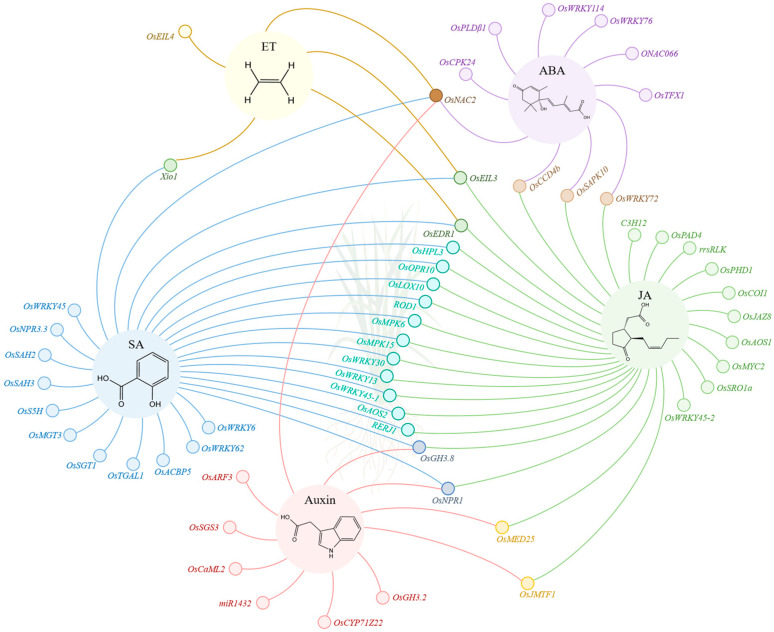
Phytohormone-mediated BLB resistance genes and the crosstalk network between phytohormones. ABA: abscisic acid; ET: ethylene; JA: jasmonic acid; SA: salicylic acid.

**Figure 3 plants-13-02541-f003:**
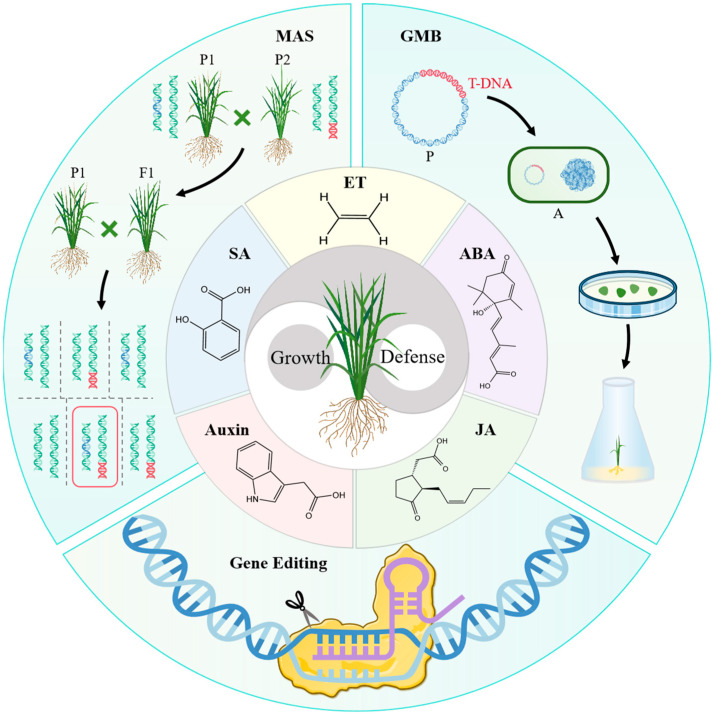
Phytohormones regulatory network in response to BLB combined with modern technology for breeding strategy. MAS: marker-assisted selection; GMB: genetically modified breeding; ABA: abscisic acid; ET: ethylene; JA: jasmonic acid; SA: salicylic acid; P: plasmid; A: agrobacterium.

**Table 1 plants-13-02541-t001:** SA-mediated BLB resistance and susceptible genes.

Gene	Gene Product	Phytohormone	Function Characteristic	Regulation Mode	Reference
*OsNPR1*	Non-expresser of pathogenesis-related 1 protein	SA, JA, Auxin	Binds to SA, then interacts with related TGA TF	Positive	[24,27]
*OsNPR3.3*	Non-expresser of pathogenesis-related 3.3 protein	SA	Participates in the *xa5*-mediated immune response, and interacts with TGAL11	Positive	[28]
*OsSAH2*	SA hydroxylase	SA	Negatively regulates the SA signaling pathway and BLB resistance	Negative	[30]
*OsSAH3*	SA hydroxylase	SA	Negatively regulates the SA signaling pathway and BLB resistance	Negative	[30]
*OsS5H*	SA hydroxylase	SA	Negatively regulates SA level and BLB resistance	Negative	[31]
*OsMGT3*	Chloroplast-localized Mg2+ transporter	SA	Negatively regulates SA level and BLB resistance	Negative	[32]
*OsSGT1*	SA glucosyltransferase	SA	Activated by *OsTGAL1* to negatively regulate resistance to *Xoo*	Negative	[34]
*OsTGAL1*	bZIP transcription factor	SA	Negatively regulates resistance to *Xoo* by activating the effect of *OsSGT1* on SA metabolism	Negative	[34]
*OsACBP5*	Acyl-CoA-binding protein	SA	Resistance is generated through an SA-induced defense response	Positive	[35]
*ROD1*	Resistance of rice to diseases 1	SA, JA	Degrades ROS to inhibit the defense response	Negative	[36]
*OsMPK6*	Mitogen-activated protein kinase	SA, JA	Regulates ROS level and the gene expression of JA and SA signaling pathways	Negative	[38,39]
*OsMPK15*	Mitogen-activated protein kinase	SA, JA	Regulates ROS level and the gene expression of JA and SA signaling pathways	Negative	[40]
*OsNAC2*	NAC transcription factor	SA, ABA, ET, Auxin	Inhibits SA signal transduction and interacts with OsEREBP1	Negative	[43,44,45,46]
*OsWRKY45-1*	WRKY transcription factor	SA, JA	Regulates SA and JA levels and the expression of defense-related genes	Negative	[4]
*OsWRKY45*	WRKY transcription factor	SA	Plays a positive role in BTH-induced BLB resistance and confers extremely strong resistance to BLB when overexpressed	Positive	[49]
*OsWRKY62*	WRKY transcription factor	SA	Negatively regulates *xa21*-mediated resistance to BLB	Negative	[50,51]
*OsWRKY6*	WRKY transcription factor	SA	Activates putative SA synthase OsICS1 to activate the SA signaling pathway	Positive	[52]
*OsWRKY13*	WRKY transcription factor	SA, JA	Activate SA signaling pathway and inhibits JA synthesis	Positive	[53]

**Table 3 plants-13-02541-t003:** ET-mediated BLB resistance and susceptible genes.

Gene	Gene Product	Phytohormone	Function Characteristic	Regulation Mode	Reference
*Xio1*	*Oryza*-specific orphan protein	ET, SA	Produces ROS and H_2_O_2_ to participate in the defense response	Positive	[94]
*OsEIL4*	Ethylene-insensitive3-like4	ET	Positively regulates the expression of ET synthesis-related genes and defense-related genes	Positive	[96]
*OsEDR1*	Mitogen-activated protein kinase	ET, SA, JA	Induces ACC and ACS to promote ET accumulation and inhibit SA and JA levels	Negative	[97]

**Table 4 plants-13-02541-t004:** Auxin-mediated BLB resistance and susceptible genes.

Gene	Gene Product	Phytohormone	Function Characteristic	Regulation Mode	Reference
*OsGH3.8*	IAA-amido synthetase	Auxin, SA, JA	Prevents the accumulation of IAA and inhibits the expression of expansins	Positive	[99]
*OsGH3.2*	IAA-amido synthetase	Auxin	Prevents the accumulation of IAA and inhibits the expression of expansins	Positive	[15]
*OsCYP71Z22*	Cytochrome P450	Auxin	Prevents the accumulation of IAA and inhibits the expression of expansins	Positive	[105,106,107]
*miR1432*	microRNA	Auxin	Disease resistance is mediated by targeting the negative regulatory gene *OsCaML2*	Positive	[112,113]
*OsCaML2*	Calmodulin-like protein 2	Auxin	Co-mediates resistance as a target of *miR1432*	Negative	[113]
*OsSGS3*	Suppressor of gene silencing3	Auxin	Regulates resistance through the OsSGS3-tasiRNA- OsARF3 module	Negative	[114]
*OsARF3*	Auxin response factors	Auxin	Regulates resistance through the OsSGS3-tasiRNA- OsARF3 module	Positive	[114]

**Table 5 plants-13-02541-t005:** ABA-mediated BLB resistance and susceptible genes.

Gene	Gene Product	Phytohormone	Function Characteristic	Regulation Mode	Reference
*OsPLDβ1*	Phospholipase D	ABA	Regulates ROS levels and the expression of defense-related genes	Negative	[122]
*OsCPK24*	Calcium-dependent protein kinase 24	ABA	Ethylicin activates OsCPK24 by influencing the ABA signaling pathway, thereby regulating rice resistance to BLB	Positive	[124]
*OsWRKY114*	WRKY transcription factor	ABA	Promotes the SA signaling pathway and downregulates the expression of ABA-related genes	Positive	[126]
*OsWRKY76*	WRKY transcription factor	ABA	Regulates defense-related genes and phytoalexin levels to regulate resistance	Negative	[50,127,128]
*ONAC066*	NAC transcription factor	ABA	Exerts its functions in disease resistance by modulating the ABA signaling pathway, sugar, and amino acid accumulations in rice	Positive	[129]
*OsTFX1*	bZIP transcription factor	ABA	The induction of *OsTFX1* depends on the type III effector gene *pthXo6*	Negative	[130,131]

## Data Availability

The original contributions presented in the study are included in the article, further inquiries can be directed to the corresponding author.

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
