# Peer review of "Mechanism of Rice Resistance to Bacterial Leaf Blight via Phytohormones"

_plants, 2024, doi:10.3390/plants13182541_

Round 1
Reviewer 1 Report
Comments and Suggestions for Authors
Major comments
The authors have collected a lot of information regarding rice resistance to bacterial leaf blight mediated by phytohormones, which is valuable. However, as mentioned in the minor comments, there are numerous inaccuracies that are concerning. Since not all issues could be covered in the minor comments, it is strongly recommended that the authors carefully review the cited references throughout the manuscript and revise to ensure that accurate information is provided.
Minor comments
Title: ‘Mechanism of Phytohormones in Rice Resistance to Bacterial Leaf Blight’ should be ‘Mechanism of Rice Resistance to Bacterial Leaf Blight via Phytohormones’.
L24: ‘insects’ is enough instead of ’Nilaparvata lugens, Cnaphalocrocis medinalis and Chilo suppressalis’.
L32&34: ‘bactericides’ would be better than ‘pesticides’.
L32: ‘but its targeted BLB is not strong, the control effect is poor,’ should be changed to ‘but its effect to the targeted BLB is not strong,’.
L34: change ‘plants’ to ‘targeted bacteria’.
L70, 79, throughout: "Mechanism of Phytohormones" should be rephrased to "Role of Phytohormones".
L95 ‘… is a classic representative of phytohormones involved in defense response [23].’
NPR1 is not a phytohormone.
L99 ‘it can increase the sensitivity to rice pests and blast[26]’: I did not find such data in the reference.
L100: insert disease in front of ‘resistance gene’ because it does not increase pest resistance (reference 24).
L102-103:’were also induced ….’ What is the subject? Show the reference. I was not able to find such data (concerning induction of OsNPR3.3) in reference 28 and 29.
L117: ’resistant’ should be changed to ‘involved in resistance’.
L118: ‘lesion phenotype’ should be ‘lesion mimic phenotype’.
L120: which data in the reference 35 and 36 show interaction between OsHPLH3 and AOS?
L140: ‘susceptibility and yield to Xoo’ should be ‘susceptibility to Xoo and yield’.
L166: ‘OsICS1 is a key enzyme ……’ would not be correct. Read the recent paper (https://doi.org/10.1111/nph.18842).
L173-174: show the reference.
L192: delete ‘(Table 2)’ as references 64 to 66 are not included in Table 2. Instead, insert ‘(Table 2)’ in the appropriate place.
Figure 1. Please confirm whether schematic drawing of ‘water hole’ is correct. I think it shows stoma and not water hole.
Table 1.
‘Non-expresser of pathogenesis-related gene’ should be ‘Non-expresser of pathogenesis-related 1 protein’. ‘Non-expresser of pathogenesis-related gene’ should be ‘Non-expresser of pathogenesis-related 3.3 protein’.
OsHPL3, OsLOX10, OsOPR10 and OsWRKY30 would be moved to Table 2 because their functions are mainly mediated by JA signal.
OsWRKY30 does not promote SA accumulation according to reference 52.
OsWRKY62: change ‘As a negative regulator of’ to ‘Negatively regulates’.
OsWRKY6: add putative in front of ‘SA synthase’.
Table 2.
OsPHD1: ‘upregulates defense-related genes’ and negative regulation mode are contradictory.
OsWRKY72: ‘susceptibility to rice’ would be ‘susceptibility to Xoo’.
Table 3.
OsEIL4: ‘Ethylene-insensitive 3’ would be ‘Ethylene-insensitive3-like4’
Figure 3.
Write what MAS, GMB, P, and A abbreviate in the legend.
Comments on the Quality of English LanguageEnglish is difficult to read and needs proofreading.
Author Response
Comments 1: Title: ‘Mechanism of Phytohormones in Rice Resistance to Bacterial Leaf Blight’ should be ‘Mechanism of Rice Resistance to Bacterial Leaf Blight via Phytohormones’.
Response 1: Thank you for your suggestion. We have revised the title to better reflect the theme of the MS. (Line 2-3)
Comments 2: L24: ‘insects’ is enough instead of ‘Nilaparvata lugens, Cnaphalocrocis medinalis and Chilo suppressalis’.
Response 2: Thank you for your reminder. This issue was also raised by another reviewer, and we apologize for the miscommunication. We have revised this sentence in the MS to eliminate any further objections. (Line 25)
Comments 3: L32&34: ‘bactericides’ would be better than ‘pesticides’.
Response 3: Thank you for your valuable feedback. We have changed “pesticides” to “bactericides”. (Line 36)
Comments 4: L32: ‘but its targeted BLB is not strong, the control effect is poor,’ should be changed to ‘but its effect to the targeted BLB is not strong,’.
Response 4: Thank you for your valuable feedback. We have changed “but its targeted BLB is not strong, the control effect is poor” to “but its effect to the targeted BLB is not strong”. (Line 34)
Comments 5: L34: change ‘plants’ to ‘targeted bacteria’.
Response 5: Thank you for your suggestion. We have changed “plants” to “targeted bacteria”. (Line 36)
Comments 6: L70, 79, throughout: "Mechanism of Phytohormones" should be rephrased to "Role of Phytohormones".
Response 6: Thank you for this suggestion. We have changed all instances of “Mechanism of Phytohormones” to "Role of Phytohormones" where necessary. (Line 71 and any parts in the MS that need to be revised)
Comments 7: L95 ‘… is a classic representative of phytohormones involved in defense response [23].’
NPR1 is not a phytohormone.
Response 7: Thank you for your reminder. We have revised this sentence to make it reasonable and coherent. Once again, we apologize for our expression error. (Line 97-98)
Comments 8: L99 ‘it can increase the sensitivity to rice pests and blast[26]’: I did not find such data in the reference.
Response 8: Thank you very much for pointing out this issue. The reference describes how enhancing rice resistance to rice blast disease increases susceptibility to pests, but we had misunderstood its meaning. We have deleted the incorrect statement. (Line 101)
Comments 9: L100: insert disease in front of ‘resistance gene’ because it does not increase pest resistance (reference 24).
Response 9: Thank you for pointing out our oversight in this detail. We have made the necessary corrections. (Line 103)
Comments 10: L102-103:’were also induced ….’ What is the subject? Show the reference. I was not able to find such data (concerning induction of OsNPR3.3) in reference 28 and 29.
Response 10: Thank you very much for your careful reading and for pointing out this error. Firstly, the issue in the first sentence was due to a word choice error, which we have now corrected. (Line 105) Secondly, the content regarding the induction of OsNPR3.3 is sourced from reference [24]. Due to our oversight, this citation was omitted, but we have now added the appropriate reference. (Line 108)
Comments 11: L117: ’resistant’ should be changed to ‘involved in resistance’.
Response 11: Thank you for your reminder. We have changed “resistant” to “involved in resistance” making this sentence more reasonable. (Line 121)
Comments 12: L118: ‘lesion phenotype’ should be ‘lesion mimic phenotype’.
Response 12: Thank you for your suggestion. We have changed “lesion phenotype” to “lesion mimic phenotype”. (Line 123)
Comments 13: L120: which data in the reference 35 and 36 show interaction between OsHPLH3 and AOS?
Response 13: Thank you for raising this issue, which helped us identify our incorrect statement. The reference discusses two branches that exhibit crosstalk, and we have revised the MS to reflect the correct interpretation. Thank you again for your careful review. (Line 125-127)
Comments 14: L140: ‘susceptibility and yield to Xoo’ should be ‘susceptibility to Xoo and yield’.
Response 14: Thank you for your suggestion. We have changed “susceptibility and yield to Xoo” to “susceptibility to Xoo and yield”. (Line 146-147)
Comments 15: L166: ‘OsICS1 is a key enzyme ……’ would not be correct. Read the recent paper (https://doi.org/10.1111/nph.18842).
Response 15: Thank you very much for your careful review. We have corrected this statement in the MS by integrating findings from two references (Choi et al. 2015; https://doi.org/10.1111/nph.18842). Although OsICS1 is not the key enzyme for SA synthesis in rice, there is evidence suggesting its involvement in SA synthesis. Therefore, we have revised the statement accordingly to “OsICS1 participates in the synthesis of SA…”. If this correction is still inaccurate, we would appreciate your further guidance for additional revisions. (Line 174)
Comments 16: L173-174: show the reference.
Response 16: Thank you very much for your reminder. Here, we have added the corresponding reference and also enriched the content of this section. (Line 176-180)
Comments 17: L192: delete ‘(Table 2)’ as references 64 to 66 are not included in Table 2. Instead, insert ‘(Table 2)’ in the appropriate place.
Response 17: Thank you for your reminder. We have moved “(Table 2)” to the appropriate position. (Line 267)
Comments 18: Figure 1. Please confirm whether schematic drawing of ‘water hole’ is correct. I think it shows stoma and not water hole.
Response 18: Thank you very much for pointing out this issue. Water hole is modified stomata surrounded by two guard cells on the epidermis, and these guard cells have generally lost the ability to control opening and closing. In our original figure, the guard cells were shown as closed, which indeed could mislead one to think they were stomata instead of water hole. We have corrected this detail and used different colors to distinguish between stomata and water hole, ensuring there is no further confusion. (Line 66-67)
Comments 19: Table 1.
‘Non-expresser of pathogenesis-related gene’ should be ’Non-expresser of pathogenesis-related 1 protein’. ‘Non-expresser of pathogenesis-related gene’ should be ‘Non-expresser of pathogenesis-related 3.3 protein’.
Response 19: Thank you for your reminder. We have changed “Non-expresser of pathogenesis-related gene” and “Non-expresser of pathogenesis-related gene” to “Non-expresser of pathogenesis-related 1 protein” and “on-expresser of pathogenesis-related 3.3 protein”. (Table 1)
Comments 20: OsHPL3, OsLOX10, OsOPR10 and OsWRKY30 would be moved to Table 2 because their functions are mainly mediated by JA signal.
Response 20: Thank you for your feedback. These four genes are indeed mainly related to JA signal, and we have moved them to Table 2. (Table 2)
Comments 21: OsWRKY30 does not promote SA accumulation according to reference 52.
Response 21: Thank you for pointing out this issue. This was due to our misunderstanding of the reference. The transcription product of OsWRKY30 can rapidly accumulate after SA and JA treatments, but OsWRKY30 only promotes the accumulation of JA. We have made the necessary revisions in the main text and Table 2 of the MS. (Line 168 and Table 2)
Comments 22: OsWRKY62: change ‘As a negative regulator of’ to ‘Negatively regulates’.
Response 22: Thank you for your suggestion. We have changed “As a negative regulator of” to “Negatively regulates”. (Table 1)
Comments 23: OsWRKY6: add putative in front of ‘SA synthase’.
Response 23: Thank you for your suggestion. We have added “putative” in front of “SA synthase”. (Table 1)
Comments 24: Table 2.
OsPHD1: “upregulates defense-related genes” and negative regulation mode are contradictory.
Response 24: Thank you very much for pointing out that this statement is contradictory. We realized that our expression was incorrect. What we intended to convey is that the loss of function of OsPHD1 leads to the upregulation of defense gene expression, indicating it negatively regulates rice resistance. We have revised the wording in the table to eliminate any discrepancies. (Table 2)
Comments 25: OsWRKY72: ‘susceptibility to rice’ would be ‘susceptibility to Xoo’.
Response 25: Thank you for your suggestion. We have changed “susceptibility to rice” in front of “susceptibility to Xoo”. (Table 2)
Comments 26: Table 3.
OsEIL4: ‘Ethylene-insensitive 3’ would be ‘Ethylene-insensitive3-like4’
Response 26: Thank you for your suggestion. We have changed “Ethylene-insensitive 3” in front of “Ethylene-insensitive3-like4”. (Table 3)
Comments 27: Figure 3.
Write what MAS, GMB, P, and A abbreviate in the legend.
Response 27: Thank you for pointing out this shortcoming. We have added the full names of these abbreviations in the legend of Figure 3, so that first-time readers can better understand the meaning of the figure. (Line 547-550)

Reviewer 2 Report
Comments and Suggestions for Authors
Revision of the Manuscript ID: plants-3167934 entitled “Mechanism of Phytohormones in Rice Resistance to Bacterial Leaf Blight”
The manuscript addresses the function of the most explored phytohormones in regulating rice resistance against a bacterial disease. Furthermore, this review tackles the molecular mechanisms behind these phytohormones' regulation and their network regarding BLB resistance in rice. The content is generally rich. The topic is not very novel, but important nonetheless.
However, the manuscript has small weaknesses, and I have some concerns prior to accepting this manuscript to Plants. The authors must address the comments and or justify some putative limitations. In detail below.
a) Title: the title is adequate to the work
b) Keywords: 5 keywords are used. keywords are fine, they are specific to the work, however, I recommend changing the 3 keywords that use the same words present in the title to expand the search results of this work in the future
c) Abstract: abstract is well-written and addresses the different parts of the work, with a clear identification of the objectives proposed for this review. Nonetheless, I suggest that the authors refrain from using abbreviations in the abstract (such as BLB, ABA, ET, etc.) since this is the first time that some readers will be exposed to that information, and may not know what those abbreviations signify.
d) Abbreviations: Ok
e) Introduction: the introduction is overall well written with small English corrections needed. It addresses the state of art of the problem suggested by the authors. The objectives are clear, which I find a good aspect. Furthermore, some suggestions should be addressed by the authors.
Lines 24-25: Please rephrase it, so it is easier to understand that the species presented there are plant pests. For example: such as plant viruses, bacteria, fungi, and some plants pests (Nilaparvata lugens,…)
The quality of Figure 1 should be improved. The words on the figure are a little bit pixelated.
Line 35: change “difficult” to “difficulties”
f) Mechanism of Phytohormones in Resistance to BLB in Rice: This section is overall well written. Moreover, this section explores with detail the influence of each phytohormone on rice resistance to BLB. My suggestion is that in the caption of Figure 2, the authors explain the meaning of the abbreviations present in the figure.
g) Mutual Regulatory Mechanism of Phytohormones in Resistance to BLB in Rice: This subsection overall is well written. The information present here is pertinent, however, it can be a bit difficult to follow the diverse regulatory mechanisms that can occur, so my suggestion is for the authors to try to include a schematic displaying the most relevant mutual regulatory mechanisms.
h) Application of Modern Breeding Techniques in Cultivating Rice Resistant to BLB: The first paragraph is lacking proper references to support the ideas present there, please add some references. Furthermore, Lines 491-493 also need proper references.
i) Summary and Prospects: It is also well written, and very clear. My suggestion is that in the caption of Figure 3, the authors explain the meaning of the abbreviations present in the figure.
j) Figures and tables: overall ok.
k) References: references are adequate to the work, and I did find an adequate use of self-references, which I find a positive aspect of the manuscript. Of the total of 144 citations, 92 are from the last ten years, which I also find adequate and another good aspect of the manuscript.
In conclusion, this manuscript can be accepted after minor revisions, the authors should address these smaller recommendations.
Comments on the Quality of English LanguageLanguage is overall fine
Author Response
|
Comments 1: 5 keywords are used. keywords are fine, they are specific to the work, however, I recommend changing the 3 keywords that use the same words present in the title to expand the search results of this work in the future |
|
Response 1:Thanks for your suggestion. We have changed some keywords that are the same as the title to expand the search results of this work. Thank you very much for your valuable comments once again. (Line 20).
|
|
Comments 2: Abstract is well-written and addresses the different parts of the work, with a clear identification of the objectives proposed for this review. Nonetheless, I suggest that the authors refrain from using abbreviations in the abstract (such as BLB, ABA, ET, etc.) since this is the first time that some readers will be exposed to that information, and may not know what those abbreviations signify. |
|
Response 2: Thanks for your suggestion. We have included the full names of these abbreviations in the abstract so that first-time readers can understand the meaning of these words. (Line 10, 13-14)
Comments 3: Lines 24-25: Please rephrase it, so it is easier to understand that the species presented there are plant pests. For example: such as plant viruses, bacteria, fungi, and some plants pests (Nilaparvata lugens,…) Response 3: Thank you for your reminder. This issue was also raised by another reviewer, and we apologize for the miscommunication. We have revised this sentence in the MS to eliminate any further objections. (Line 25)
Comments 4: The quality of Figure 1 should be improved. The words on the figure are a little bit pixelated. Response 4: Thank you for your valuable comment. We adjusted the fonts in Figure 1 to make them appear clearer. (Line 66-67)
Comments 5: Line 35: change “difficult” to “difficulties” Response 5: Thanks for your careful reading. We have changed “difficult” to ”difficulties”. (Line 37)
Comments 6: Mechanism of Phytohormones in Resistance to BLB in Rice: This section is overall well written. Moreover, this section explores with detail the influence of each phytohormone on rice resistance to BLB. My suggestion is that in the caption of Figure 2, the authors explain the meaning of the abbreviations present in the figure. Response 6: Thank you for your reminder. We have explain the meaning of the abbreviations present in the figure 2. (Line 78-79)
Comments 7: Mutual Regulatory Mechanism of Phytohormones in Resistance to BLB in Rice: This subsection overall is well written. The information present here is pertinent, however, it can be a bit difficult to follow the diverse regulatory mechanisms that can occur, so my suggestion is for the authors to try to include a schematic displaying the most relevant mutual regulatory mechanisms. Response 7: Thank you very much for your valuable feedback. Regarding the content on phytohormones crosstalk, we have already illustrated this regulatory network in Figure 2. Additionally, we have added annotations in the caption of Figure 2 to indicate that the figure also depicts phytohormones crosstalk. We have also included a reference to Figure 2 in the corresponding subsection of the main text. Finally, thank you again for this suggestion, which has helped make our MS more complete. (Line 78-79, 477)
Comments 8: Application of Modern Breeding Techniques in Cultivating Rice Resistant to BLB: The first paragraph is lacking proper references to support the ideas present there, please add some references. Furthermore, Lines 491-493 also need proper references. Response 8: Thank you very much for your feedback. We have added the corresponding references in the first paragraph of this subsection and in Line 491-493 to make it more convincing. (Line 482-490)
Comments 9: Summary and Prospects: It is also well written, and very clear. My suggestion is that in the caption of Figure 3, the authors explain the meaning of the abbreviations present in the figure. Response 9: Thank you for your reminder. We have explain the meaning of the abbreviations present in the figure 3. (Line 548-550) |

Reviewer 3 Report
Comments and Suggestions for Authors
The present paper is devoted to the study of the mechanisms of resistance to BLB in rice induced by different phytohormones. The topic is very interesting and the present MS is trying to cover many aspects of this problem. However, there are issues that require the authors’ attention. A particularly big issue is that due to the deficiencies in the language a lot of sentences lose their meaning.
1. Abstract.
The abbreviations should be done for “BLB” and “Xoo”.
2. Although, overall the language is not poor, the text needs to be inspected by the English proofreader. Some phrases are awkward.
For instance,
P. 1. The title
“Mechanism of Phytohormones in Rice Resistance to Bacterial Leaf Blight”
P. 3, L. 70. 2. “Mechanism of Phytohormones in Resistance to BLB in Rice”
Mechanism of what?
In many cases some phrases lose their sense.
For instance,
P. 3, L. 93-96. “Many genes that respond to BLB regulate their resistance by regulating SA levels or their signal transduction.”
P. 4, L. 100. “The SA signal mediating its resistance is involved in the xa5-mediated immune response of rice against BLB, and this SA signal is independent of OsNPR1[28, 29].”
Which resistance???
“NPR1 is a key regulator of systemic acquired resistance (SAR) in Arabidopsis and rice, and is a classic representative of phytohormones involved in defense response[23].”
NPR1 is a classic representative of phytohormones???
P. 4, L. 116-117. “All of the above genes are resistant to Xoo through SA.”
What does it mean??
Through the text there are many similar nonsense phrases and other inaccuracies that need to be corrected. I encourage the authors to double check the entire MS for this type of errors.
Sometimes there is a lack of certain words in the phrases that is why the meaning of the phrases is lost.
For instance,
P. 11, L. 288-289. “The EIN3/EILs family is at the hub of ET signaling pathway and plays an important role in responding to plant development and stress[93].”
What is the family?
P. 12, L. 337-338. “In addition to the GH3 gene family, many studies have demonstrated that IAA is involved in regulating the resistance of rice to BLB.”
There is no linking word that’s why it is nonsense.
8. P. 14, L. 383-384. “Known studies have shown that pathogens often mediate the defense responses involved in ABA…”
…involved in ABA?
P. 11, L. 292-295. “The regulation of ET between JA and SA is an important feature of phytohormone signaling regulation in rice…”
This phrase is nonsense.
There are mistakes, inaccuracies, etc.
I encourage the authors to double check the entire text with the help of a native speaker or professional proofreader to improve the language.
3. Subsection “2.1. Mechanism of SA in Resistance to BLB in Rice”
There are a lot of sentences without mention of species. It’s boring. However it would be okay if there was mention to rice plants in the first two phrases.
4. Some clarification and deciphering of genes are necessary especially in section regarding the role of SA. It would be better if there was clarification of the role of genes in the metabolism of SA in the text.
5. The text is not consistent. It should be more concise.
For instance, P. 4, L. 130-131. “At present, studies on the involvement of SA in rice resistance to Xoo have made some progress, and there are some similarities. For example, MAPK signal transduction plays an important role in the immune response mediated by disease resistance (R) genes involved in SA[40]. ”
Why are the “similarities” here?
6. There are many parts in the sections devoted to individual phytohormones that contain the data on crosstalk between different plant hormones. There is a whole section “3. Mutual Regulatory Mechanism of Phytohormones in Resistance to BLB in Rice” which may overlap with these parts.
7. The section “Application of Modern Breeding Techniques in Cultivating Rice Resistant to BLB” is interesting but it doesn't quite fit into the overall plot of MS. It would be better if it contains some data on the role of plant hormones or, for example, the manipulation with genes related to the metabolism or signaling of phytohormones.
Comments on the Quality of English LanguageAlthough, overall the language is not poor, the text needs to be inspected by the English proofreader. Due to the deficiencies in the language the meaning of some sentences lose their meaning. I encourage the authors to double check the text to improve the quality of language.
Author Response
|
Comments 1: Abstract. The abbreviations should be done for “BLB” and “Xoo”. Response 1: Thanks for your suggestions. We have included the full names of these abbreviations in the abstract so that first-time readers can understand the meaning of these words. (Line 10)
Comments 2: Some phrases are awkward. For instance,
“Mechanism of Phytohormones in Rice Resistance to Bacterial Leaf Blight”
Mechanism of what? Response 2: Thank you for pointing out this shortcoming to us, we are very sorry for our contradictory expression. We have revised the expression of the title and subheadings respectively. (Line 2, 71)
Comments 3: In many cases some phrases lose their sense. For instance,
Which resistance??? Response 3: Thank you for your careful reading, we feel so sorry for our misrepresentation. We want to express here is the improved resistance of rice, and corresponding modifications have been made in the MS. (Line 95, 107)
Comments 4: “NPR1 is a key regulator of systemic acquired resistance (SAR) in Arabidopsis and rice, and is a classic representative of phytohormones involved in defense response[23].” NPR1 is a classic representative of phytohormones??? Response 4: Thank you for your reminder. We have revised this sentence to make it reasonable and coherent. Once again, we apologize for our expression error. (Line 97-98)
Comments 5: P. 4, L. 116-117. “All of the above genes are resistant to Xoo through SA.” What does it mean?? Response 5: Thank you for pointing out the issue with this sentence. We have made the necessary modifications in the MS to ensure it conveys the correct meaning. (Line 121)
Comments 6: Sometimes there is a lack of certain words in the phrases that is why the meaning of the phrases is lost. For instance,
What is the family? Response 6: Thank you for your reminder. We are referring to the EIL3 transcription factors family here. We have supplemented the specific wording in the MS. Thank you once again for your careful review. (Line 298) |
|
Comments 7: P. 12, L. 337-338. “In addition to the GH3 gene family, many studies have demonstrated that IAA is involved in regulating the resistance of rice to BLB.” There is no linking word that’s why it is nonsense. |
|
Response 7: Thank you for pointing out our mistake in this content. At this point, we have added appropriate conjunctions to make this sentence more logical and coherent. (Line 349)
|
|
Comments 8: P. 14, L. 383-384. “Known studies have shown that pathogens often mediate the defense responses involved in ABA…” …involved in ABA? |
|
Response 8: Thank you for your reminder. We have modified certain words in this sentence to achieve the intended expression effect. (Line 397)
Comments 9: P. 11, L. 292-295. “The regulation of ET between JA and SA is an important feature of phytohormone signaling regulation in rice…” This phrase is nonsense. Response 9: Thanks for your careful reading and valuable comments. In response to this issue, we have revised this sentence in the MS to ensure it is unambiguous and coherent. (Line 303-304)
Comments 10: Subsection “2.1. Mechanism of SA in Resistance to BLB in Rice” There are a lot of sentences without mention of species. It’s boring. However it would be okay if there was mention to rice plants in the first two phrases. Response 10: Thanks for your valuable suggestions. Taking your suggestions into account, we have included a reference to the rice species here to make this section more coherent. (Line 83)
Comments 11: Some clarification and deciphering of genes are necessary especially in section regarding the role of SA. It would be better if there was clarification of the role of genes in the metabolism of SA in the text. Response 11: Thank you very much for this valuable suggestion. We have enriched the MS by elaborating on the roles of some SA-mediated resistance genes in SA metabolism, making the content more complete. (Line 108-109, 117-118, 176, 185)
Comments 12: The text is not consistent. It should be more concise. For instance, P. 4, L. 130-131. “At present, studies on the involvement of SA in rice resistance to Xoo have made some progress, and there are some similarities. For example, MAPK signal transduction plays an important role in the immune response mediated by disease resistance (R) genes involved in SA[40]. ” Why are the “similarities” here? Response 12: Thank you for pointing out this issue. We have rephrased it here to achieve the intended expression effect. (Line 138)
Comments 13: There are many parts in the sections devoted to individual phytohormones that contain the data on crosstalk between different plant hormones. There is a whole section “3. Mutual Regulatory Mechanism of Phytohormones in Resistance to BLB in Rice” which may overlap with these parts. Response 13: Thank you very much for pointing out this issue. Regarding the crosstalk of phytohormones, we have mentioned it in the previous sections where the roles of each phytohormone in BLB are discussed. However, the main summary is in the section titled 'Mutual Regulatory Mechanism of Phytohormones in Rice Resistance to BLB.' The earlier sections focus on the crosstalk involving individual phytohormone, while this section outlines the crosstalk network among the five phytohormones. Although there is some overlapping content, the emphasis is different. Additionally, we have reduced the crosstalk content in the earlier sections to make the MS more concise.
Comments 14: The section “Application of Modern Breeding Techniques in Cultivating Rice Resistant to BLB” is interesting but it doesn't quite fit into the overall plot of MS. It would be better if it contains some data on the role of plant hormones or, for example, the manipulation with genes related to the metabolism or signaling of phytohormones. Response 14: Thank you very much for your insightful suggestion. In this section, we have added content on the role of modern breeding techniques in phytohormones-mediated BLB resistance, making the section “Application of Modern Breeding Techniques in Cultivating Rice Resistant to BLB” more convincing and more relevant to the overall MS. (Line 511-512, 522-525) |

Round 2
Reviewer 1 Report
Comments and Suggestions for Authors
Major comments
In the previous major comment, I wrote ‘it is strongly recommended that the authors carefully review the cited references throughout the manuscript and revise to ensure that accurate information is provided.’ However, it is regrettable that authors did not follow the comment and I have to point out easy mistakes again. In addition, there are too many English errors to point out. The authors need to carefully review the entire text.
L296 to 301: reference 93 is related to ethylicin but not ethylene.
Minor comments
L10: ‘Oryzae’ should be ‘oryzae’ and italic.
L132: ‘the activation role’ should be ‘the activation mechanism’.
L125 to 136: move to section 2.2.
L164: ‘OsWRKY45-1 and OsWRKY45-2’ should be ‘OsWRKY45-1(from Japonica rice) and OsWRKY45-2 (from indica rice)’.
L168: ‘its resistance to Xoo’ is not correct because OsWRKY45-1 is a negative regulator of Xoo resistance. Knockout of OsWRKY45-1 increases with the accumulation.
L168: Another paper (DOI: 10.1111/J.1364-3703.2011.00732.X) shows that WRKY45 (from Japonica rice) is a positive regulator of of Xoo resistance. Authors should also introduce this paper in the text and Table 1.
L169 to 174: move to section 2.2.
Table 1 caption: ‘SA-mediated BLB resistance genes’ would be ‘SA-mediated BLB resistance and susceptible genes’. Captions of Table 2 to 5 should be changed similar.
Table 1 OsWRKY62: delete ‘of’.
L207: ‘BLB’ would be changed to ‘disease’ or ‘Xanthomonas’.
L208: is BLB OK (in maize)?
L218: ROS is not a negative regulator of Xoo resistance. Change the sentence.
L219: loss-function would be loss-of-function, throughout the manuscript.
L280 ‘weaken resistance to biotroph Xoo and confer resistance’: it is a contradictory statement.
Table2 OsJAZ8: ‘Negative’ would be ‘Negatively’.
L439: add ‘resistance’ behind ‘BLB’.
L506: ‘susceptible’ would be ‘resistant’.
L535: ‘it is very important to’ should be deleted.
Comments on the Quality of English LanguageThere are too many English errors to point out.
Author Response
Comments 1: Title: In the previous major comment, I wrote ‘it is strongly recommended that the authors carefully review the cited references throughout the manuscript and revise to ensure that accurate information is provided.’ However, it is regrettable that authors did not follow the comment and I have to point out easy mistakes again. In addition, there are too many English errors to point out. The authors need to carefully review the entire text.
Response 1: Thank you very much for pointing out our shortcomings in this regard. In response to this issue, we have reviewed and revised the entire MS, correcting expressions that were in dispute. If there are still any errors we haven't corrected, please bring them to our attention, and we will do our best to make the necessary revisions.
Comments 2: L296 to 301: reference 93 is related to ethylicin but not ethylene.
Response 2: Thank you for pointing out this significant error. We sincerely apologize for our oversight. We have made corrections in the MS, removing this content from section 2.3 and Table 3, and adjusted Figure 2 accordingly. Additionally, we have refined this content in section 2.5 and Table 5. (Figure 2, Table 3, Table 5, Line 411-415)
Comments 3: L10: ‘Oryzae’ should be ‘oryzae’ and italic.
Response 3: Thank you for this reminder. We have changed “Oryzae” to “oryzae”. (Line 10)
Comments 4: L132: ‘the activation role’ should be ‘the activation mechanism’.
Response 4: Thank you for your suggestion. We have changed “the activation role” to “the activation mechanism”. (Line 201)
Comments 5: L125 to 136: move to section 2.2.
Response 5: Thank you for your suggestion. We have moved the content regarding JA-related genes from section 2.1 to section 2.2. (Line 195-206)
Comments 6: L164: ‘OsWRKY45-1 and OsWRKY45-2’ should be ‘OsWRKY45-1(from Japonica rice) and OsWRKY45-2 (from indica rice)’.
Response 6: Thank you for this reminder. We have changed “OsWRKY45-1 and OsWRKY45-2" to “OsWRKY45-1(from Japonica rice) and OsWRKY45-2 (from indica rice)”. (Line 149)
Comments 7: L168: ‘its resistance to Xoo’ is not correct because OsWRKY45-1 is a negative regulator of Xoo resistance. Knockout of OsWRKY45-1 increases with the accumulation.
Response 7: Thank you very much for your careful reading and for pointing out this error. We revised the wording in the MS to clarify that the resistance originates from the OsWRKY45-1 knockout lines. (Line 153)
Comments 8: L168: Another paper (DOI: 10.1111/J.1364-3703.2011.00732.X) shows that WRKY45 (from Japonica rice) is a positive regulator of of Xoo resistance. Authors should also introduce this paper in the text and Table 1.
Response 8: Thank you very much for your suggestion. We have added content related to OsWRKY45 in the MS and Table 1, and analyzed the differences between OsWRKY45 and OsWRKY45-1. (Line 155-159, Table 1)
Comments 9: L169 to 174: move to section 2.2.
Response 9: Thank you for your suggestion. We have moved the content regarding JA-related genes from section 2.1 to section 2.2. (Line 206-212)
Comments 10: Table 1 caption: ‘SA-mediated BLB resistance genes’ would be ‘SA-mediated BLB resistance and susceptible genes’. Captions of Table 2 to 5 should be changed similar.
Response 10: Thank you for your suggestion. We have accordingly revised the captions of all the tables . (Table 1-5)
Comments 11: Table 1 OsWRKY62: delete ‘of’.
Response 11: Thank you for your reminder. We have deleted “of”. (Table 1)
Comments 12: L207: ‘BLB’ would be changed to ‘disease’ or ‘Xanthomonas’.
Response 12: Thank you very much for your careful review. We have changed “BLB” to “disease”. (Line 194)
Comments 13: L208: is BLB OK (in maize)?
Response 13: Thank you for your reminder. We revised the wording in this content, changing 'BLB' to 'disease' to avoid controversy. (Line 194)
Comments 14: L218: ROS is not a negative regulator of Xoo resistance. Change the sentence.
Response 14: Thank you very much for pointing out this issue. We intended to convey that rrsRLK functions as a negative regulator of Xoo, not ROS. We apologize for the miscommunication and have revised the sentence to accurately reflect our intended meaning. (Line 222)
Comments 15: L219: loss-function would be loss-of-function, throughout the manuscript.
Response 15: Thank you very much for your reminder. We have changed “loss-function” to “loss-of-function” throughout the MS.
Comments 16: L280 ‘weaken resistance to biotroph Xoo and confer resistance’: it is a contradictory statement..
Response 16: Thank you for pointing out this mistake. We have revised this content in the MS to clarify that OsEIL3 loss-of-function mutant weakens resistance to Xoo, whereas under normal conditions, it enhances resistance. (Line 280-281)
Comments 17: Table2 OsJAZ8: ‘Negative’ would be ‘Negatively’.
Response 17: Thank you for your reminder. We have changed “Negative” to ”Negatively”. (Table 2)
Comments 18: L439: add ‘resistance’ behind ‘BLB’.
Response 18: Thank you for your suggestion. We have added “resistance” behind “BLB”. (Line 436)
Comments 19: L506: ‘susceptible’ would be ‘resistant’.
Response 19: Thank you very much for your thorough review and suggestion. However, what we intended to express here is the introduction of resistant traits into susceptible varieties with desirable characteristics through MAS. Therefore, we would like to retain the term “susceptible” . If our understanding is incorrect, please provide your suggestion for revision. (Line 502)
Comments 20: L535: ‘it is very important to’ should be deleted.
Response 20: Thank you for your feedback. We have deleted “it is very important to”. (Line 535)
|
3. Response to Comments on the Quality of English Language |
|
Point 1: There are too many English errors to point out. |
|
Response 1: Thank you very much for reviewing our MS and for your suggestions regarding the quality of the English. We have reviewed the entire MS and corrected several problematic expressions. Additionally, we have consulted native English speakers to review and revise the English in the MS. If there are still any issues with the quality of the English, please do not hesitate to point them out, and we will make every effort to correct them. |

Reviewer 3 Report
Comments and Suggestions for Authors
Although the paper has been improved, there are still deficiencies in the MS that must be resolved before publication.
1. P. 3, L. 97-98.
“Many genes that respond to BLB regulate resistance to rice by controlling SA levels or their signal transduction.”
…resistance to rice??? How genes can regulate and control? …“their signal transduction”???
2. P. 4, L. 142-143.
“For example, MAPK signal transduction plays an important role in the immune response mediated by disease resistance (R) genes involved in SA [41].”
…involved in SA???
3. P. 5, L. 169-172.
“Overexpression of OsWRKY30 showed increased resistance to pathogens, and the transcription of OsWRKY30 accumulated rapidly under the influence of SA and JA, but it specifically stimulated the endogenous accumulation of JA[53]. It also plays an important role in OsMKK3-OsMPK7-OsWRKY30[54] and OsMAPK6-OsLIC-OsWRKY30[55] Xoo resistance modules.”
Overexpression of OsWRKY30 showed increased resistance to pathogens…??? … the transcription of OsWRKY30 accumulated rapidly???
“it” – what does it refer to?? These sentences need to be corrected.
The word “it” is used a lot and it is not always clear what it means.
4. P. 11-12. L. 310-312.
The regulation of ET??? What does it mean?
5. P. 13, L. 355-356.
The sentence is awkward. Nothing has essentially changed after author’s correction.
You can just write:
“Moreover, in addition to research on GH3 gene family, many studies have demonstrated that IAA is involved in regulating the resistance of rice to BLB” – and this will be more informative.
6. P. 17. L. 513-514.
“Among them, there are already applications of BLB responses involving phytohormones in GMB, such as OsWRKY13 and OsWRKY45-2 [4, 59].”
This sentence is difficult to understand. Please rephrase it.
7. P. 17-18, L. 524-527.
“Regarding the application of gene editing in phytohormones-mediated BLB resistance, there are also some precedents. For example, using CRISPR/Cas9 techniques to knock out OsEDR1 and OsMKK15 can produce lines with enhanced resistance [44, 149].”
This phrase is not informative. What do phytohormones have to do with it? Please, expand.
8. P. 18, L. 534-536.
“Phytohormone is also one of the effective factors to resist Xoo, so it is very important to exploring its regulatory mechanism is crucial.”
I recommend to change as:
“Phytohormones effectively protect rice from the bacterial leaf blight, so it is very important to explore their regulatory mechanism in developing the resistance to Xoo.”
Again, the text needs to be inspected by qualified proofreader.
Comments on the Quality of English LanguageThe text needs to be inspected by qualified proofreader.
Author Response
|
Comments 1: P. 3, L. 97-98. “Many genes that respond to BLB regulate resistance to rice by controlling SA levels or their signal transduction.” …resistance to rice??? How genes can regulate and control? …“their signal transduction”??? |
|
Response 1: Thank you for pointing out the problematic expression in this section. In the MS, we have clarified the regulatory function of the genes and corrected the erroneous wording to ensure that it reads without ambiguity. (Line 94-95).
|
|
Comments 2: P. 4, L. 142-143. “For example, MAPK signal transduction plays an important role in the immune response mediated by disease resistance (R) genes involved in SA [41].” …involved in SA??? |
|
Response 2: Thank you for pointing out this incorrect expression. We have rephrased and revised the sentence to accurately convey our intended meaning. (Line 129)
Comments 3: P. 5, L. 169-172. “Overexpression of OsWRKY30 showed increased resistance to pathogens, and the transcription of OsWRKY30 accumulated rapidly under the influence of SA and JA, but it specifically stimulated the endogenous accumulation of JA[53]. It also plays an important role in OsMKK3-OsMPK7-OsWRKY30[54] and OsMAPK6-OsLIC-OsWRKY30[55] Xoo resistance modules.” Overexpression of OsWRKY30 showed increased resistance to pathogens…??? … the transcription of OsWRKY30 accumulated rapidly??? “it” – what does it refer to?? These sentences need to be corrected. The word “it” is used a lot and it is not always clear what it means. Response 3: Thank you for your thorough review and valuable suggestions. Apologies for the confusion in our previous expression, which led to some disagreement. We have revised and adjusted this section to ensure that it is free from grammatical issues and accurately conveys our intended meaning. Additionally, regarding the use of “it” in the MS, we have clarified all instances of ambiguous references, ensuring that they are now clear. Once again, we sincerely appreciate your valuable suggestion (Line 206-212)
Comments 4: P. 11-12. L. 310-312. The regulation of ET??? What does it mean?. Response 4: Thank you for your reminder. What we intended to convey is that the crosstalk between JA and SA is fine-tuned by ET signaling. We have revised this sentence in the MS to ensure that it clearly expresses this idea. (Line 306)
Comments 5: P. 13, L. 355-356. The sentence is awkward. Nothing has essentially changed after author’s correction. You can just write: “Moreover, in addition to research on GH3 gene family, many studies have demonstrated that IAA is involved in regulating the resistance of rice to BLB” – and this will be more informative. Response 5: Thank you for your suggestion. We have revised this sentence in the MS according to your suggestion. (Line 350-351)
Comments 6: P. 17. L. 513-514. “Among them, there are already applications of BLB responses involving phytohormones in GMB, such as OsWRKY13 and OsWRKY45-2 [4, 59].” This sentence is difficult to understand. Please rephrase it. Response 6: Thank you very much for your valuable feedback. We expanded and refined this sentence to make it easier to understand. (Line 510-512)
Comments 7: P. 17-18, L. 524-527. “Regarding the application of gene editing in phytohormones-mediated BLB resistance, there are also some precedents. For example, using CRISPR/Cas9 techniques to knock out OsEDR1 and OsMKK15 can produce lines with enhanced resistance [44, 149].” This phrase is not informative. What do phytohormones have to do with it? Please, expand. Response 7: Thank you very much for your valuable feedback. We expanded this sentence, adding details about the application of gene editing technology in demonstrating the relationship between phytohormones and rice defense, clarifying the connection among the three. (Line 522-528)
Comments 8: P. 18, L. 534-536. “Phytohormone is also one of the effective factors to resist Xoo, so it is very important to exploring its regulatory mechanism is crucial.” I recommend to change as: “Phytohormones effectively protect rice from the bacterial leaf blight, so it is very important to explore their regulatory mechanism in developing the resistance to Xoo.”. Response 8: Thank you for your suggestion. We have revised this sentence in the MS according to your suggestion. (Line 535-536)
|
|
3. Response to Comments on the Quality of English Language |
|
Point 1: The text needs to be inspected by qualified proofreader. |
|
Response 1: Thank you very much for reviewing our MS and for your suggestions regarding the quality of the English. We have reviewed the entire MS and corrected several problematic expressions. Additionally, we have consulted native English speakers to review and revise the English in the MS. If there are still any issues with the quality of the English, please do not hesitate to point them out, and we will make every effort to correct them.
|

Round 3
Reviewer 1 Report
Comments and Suggestions for Authors
The revised paper properly responded to my comments. Therefore, I do not have additional comments.